# Does small talk with a medical provider affect ChatGPT's medical counsel? Performance of ChatGPT on USMLE with and without distractions

Myriam Safrai [1,2], Amos Azaria [3]*

**1** Department of Obstetrics and Gynecology, Chaim Sheba Medical Center (Tel Hashomer), Sackler Faculty of Medicine, Tel Aviv University, Tel Aviv, Israel, **2** Department of Obstetrics, Gynecology and Reproductive Sciences, Magee-Womens Research Institute, University of Pittsburgh School of Medicine, Pittsburgh, PA, United States of America, **3** School of Computer Science, Ariel University, Ari'el, Israel

* amos.azaria@ariel.ac.il

**Data Availability Statement:** https://datadryad. org/stash/share/y1HZFsTgw4W6LaodQm5ROY6o J7M4CaEaMo0BFGBn-2I.

## Abstract

Efforts are being made to improve the time effectiveness of healthcare providers. Artificial intelligence tools can help transcript and summarize physician-patient encounters and produce medical notes and medical recommendations. However, in addition to medical information, discussion between healthcare and patients includes small talk and other information irrelevant to medical concerns. As Large Language Models (LLMs) are predictive models building their response based on the words in the prompts, there is a risk that small talk and irrelevant information may alter the response and the suggestion given. Therefore, this study aims to investigate the impact of medical data mixed with small talk on the accuracy of medical advice provided by ChatGPT. USMLE step 3 questions were used as a model for relevant medical data. We use both multiple-choice and open-ended questions. First, we gathered small talk sentences from human participants using the Mechanical Turk platform. Second, both sets of USLME questions were arranged in a pattern where each sentence from the original questions was followed by a small talk sentence. ChatGPT 3.5 and 4 were asked to answer both sets of questions with and without the small talk sentences. Finally, a board-certified physician analyzed the answers by ChatGPT and compared them to the formal correct answer. The analysis results demonstrate that the ability of ChatGPT-3.5 to answer correctly was impaired when small talk was added to medical data (66.8% vs. 56.6%; p = 0.025). Specifically, for multiple-choice questions (72.1% vs. 68.9%; p = 0.67) and for the open questions (61.5% vs. 44.3%; p = 0.01), respectively. In contrast, small talk phrases did not impair ChatGPT-4 ability in both types of questions (83.6% and 66.2%, respectively). According to these results, ChatGPT-4 seems more accurate than the earlier 3.5 version, and it appears that small talk does not impair its capability to provide medical recommendations. Our results are an important first step in understanding the potential and limitations of utilizing ChatGPT and other LLMs for physician-patient interactions, which include casual conversations.

**Funding:** Amos Azaria: Ministry of Science and Technology, Israel. No award number. https://www.gov.il/he/departments/general/most_rfp_application_guide The funders had no role in study design, data collection and analysis, decision to publish, or preparation of the manuscript.

**Competing interests:** The authors have declared that no competing interests exist.

## Introduction

One of the key, yet most time-consuming, healthcare tasks is charting and creating medical notes, a task requiring extensive time of healthcare providers [1]. In fact, this task often requires healthcare providers to spend as much, if not more time than they do in direct patient interaction [2, 3]. For example, in a survey, 67% of the residents reported spending in excess of 4 hours daily on documentation [1]. Despite the importance of medical notes [4, 5] no changes have been made to their format, besides having transferred the responsibility of writing them from other medical team members to the physicians [6]. This shift has created a burden for medical providers [7] and physician burnout [8]. Moreover, the recent implementation of electronic health records (EHRs) has significantly increased clinician documentation time [9], making it the most time-consuming physician activity [10]. This emphasizes the pressing need to improve the way of charting and making medical notes.

Large Language Models (LLMs) have been suggested as a possible solution, improving healthcare documentation, creating notes, summarizing physician-patient encounters, and even providing meaningful suggestions for further treatments [11, 12]. For example, Chat Generative Pre-trained Transformer 3.5 (ChatGPT-3.5) has been shown to generate a correct diagnosis for 93% of clinical cases with common chief complaints [13] and screening breast cancer with an average correct rate of 88.9% [14]. In addition, ChatGTP-3.5 was able to provide general medical information on common retinal disease [15], in almost every subject in genecology [16] and in cancer subjects [17]. Moreover, another article demonstrated ChatGPT-3.5's ability to generate clinical letters with high overall accuracy and humanization [18]. Recent investigations have also shown ChatGPT-3.5's ability to write medical notes [12] and to generate a discharge note based on a brief description [19]. More recently, a newer version ChatGPT-4 was released. ChatGPT-4 has the ability to process a greater word limit, a stronger ability to solve complex problems, and image recognition [20]. This version has additionally shown greater capabilities in terms of clinical evaluation [21, 22]. Namely, while ChatGPT-3.5 has obtained a score of 60.9% on a US sample clinical exam, ChatGPT-4 obtained a score of 89.8% on the same exam [21]. A similar result was obtained on the Japanese medical exam, in which ChatGPT-3.5 obtained an average score of 121.3 on the first part of the exam and 149.7 on the second part, while ChatGPT-4 obtained an average score of 167.7 on the first part of the exam and 221.5 on the second part [22].

Following the success of ChatGPT in the medical field, the technology has been tested to summarize physician-patient encounters [23]. Those appointments between healthcare providers and patients form the foundation of medical care [24]. They necessitate medical evaluations, including the provider's focus on patient needs, obtaining medical histories [25, 26], conducting physical examinations [27, 28] and performing additional tests if necessary [29, 30]. Moreover, they also entail non-medical tasks such as documenting patient records, organizing notes, and making referrals.

However, since healthcare and patient discussions are unique and based on trust, in addition to medical information, they often include small talk and other information irrelevant to medical concerns [31, 32]. Those unique exchanges are an important part of the relationship between medical providers and patients and are common among different cultures [33]. In traditional Chinese medicine doctors actively initiate small talks to acquire holistic information for diagnosis and attach significant importance to them [33]. In contrast, such interaction with small talk, has been found to alter the technical skills and performances of medical students, [34]. These controversies raise concerns regarding their potential impact on LLMs.

LLMs have the potential to streamline the note-taking process for health practitioners by automating it in two steps. Current technologies enable converting audio information, such as

conversations, into transcripts [35]. LLMs can then analyze and create medical notes, such as discharge summaries, in real-time [19]. However, the unique nature of discussions between healthcare providers and patients, which often interweave small talk with critical information, poses a challenge. Given that LLMs are predictive models that generate responses based on the input prompt [36], further investigation is needed to assess whether small talk impede the LLM's ability to formulate accurate medical notes and summaries.

Despite the growing number of studies on the potential of using AI for healthcare purposes, to the best of our knowledge, none have assessed this unique aspect of healthcare-patient content interactions and the effect that casual conversation and unrelated information could have on the efficacy of ChatGPT to process medical information and therefore be used to write medical notes summarizing physician-patient interaction. This study aims to investigate the impact of interspersing medical data with casual conversation on the precision of medical recommendations provided by ChatGPT3.5 and ChatGPT-4.

To clearly delineate the practical application of this paper, the study targets the use of LLM's alongside a practitioner for charting and summarizing health care patient interaction, including small talk, but with practitioners having significantly less manual work to accomplish. LLM's can save time and resources by automating the note-taking process of practitioners. This requires the ability of the LLM's to process relevant information. However, investigation is required to determine whether small talk inhibits the LLM in formulating accurate medical conclusions. Moreover, LLM's can suggest medical conclusions such as potential treatment and diagnosis, saving time for the practitioner. If small talk interferes with the performance of LLM's, practitioners cannot rely on them for note-taking due to 'noise' causing inaccuracies in medical records.

## Material and methods

### Medical information

To assess ChatGPT's capabilities in medical reasoning, we evaluate its responses to questions from the United States Medical Licensing Examination (USMLE). This exam has been successfully used to assess the medical logic of LLMs in previous studies [37]. Specifically, to evaluate the LLM's proficiency in addressing clinical queries, we selected the Step 3 exam, which is the final examination in the USMLE sequence that qualifies individuals to practice medicine unsupervised. The multiple-choice questions in this exam primarily test knowledge related to diagnosis and clinical management and reflect clinical situations that a general physician might encounter https://www.usmle.org/step-exams/step-3/step-3-exam-content.

USMLE Step 3 questions were sourced from the dataset provided by Kung et al. [37]. Two distinct sets of questions were utilized in the study. The first comprised the original multiple-choice (MC) questions from the USMLE Step 3 exam, while the second presented the same questions in an open-ended (OE) format. Each set contained 122 questions. This study is exempt from Institutional Review Board (IRB) review. It did not involve any interaction or intervention with human subjects nor did it access identifiable private information. Participants provided unidentifiable, non-medical, generic sentences for analysis. The primary focus of this research was ChatGPT, not human subjects. All other applicable ethical guidelines were adhered to during the conduct of this study.

### Obtaining small talk sentences

We conducted a survey on Amazon's Mechanical Turk platform, which allows researchers to recruit participants for various tasks, including online surveys and experiments. Mechanical Turk has gained considerable popularity in recent years as a tool for research due to its

efficiency, cost-effectiveness, and the ability to reach a vast pool of participants [38]. The survey was conducted from 7/24/2023 to 7/25/2023.

In our survey, we required the participants to write sentences with at least 10 words to encourage more thoughtful and meaningful responses and reduce the likelihood of individuals providing rushed, brief answers (e.g., "I ate something", "I saw someone", etc.). This is because we aim for participants to produce meaningful sentences that emulate small talk, ensuring they convey information in a casual conversational manner. The participants confirmed that they approve that the provided information will be used for research.

The participants were provided the following instructions. "Please write 5 different sentences as if you were talking to your friend. Each sentence must describe something that has happened to you or an action that you have performed in the past few days. The sentences should not depend on each-other. It is OK to write sentences about simple everyday occurrences (e.g., "1. I sat on a chair on my balcony and looked at the cars passing by."). Each sentence should be at least 10 words long."

We note that we intentionally framed the small talk in the context of "talking to a friend" rather than talking to a physician, since we did not want the small talk sentences to have any true influence on the correct answer. By framing the small talk in the context of talking to a friend, we aimed for the correct diagnosis to remain unchanged.

We elicited 35 participants, each provided 5 sentences. This resulted in 175 sentences. The following are some examples of sentences we received from the Mechanical Turk workers:

1. I had a great time catching up with my friends at the coffee shop.

2. I finished reading a great book and I'm looking for my next one.

3. I biked to the park and watched the birds for an hour.

All sentences shorter than 10 words were removed. The remaining sentences were converted to a third person's view, to better align with the USMLE format. This resulted in a list of 143 small talk sentences, which are provided in the appendix.

Converting the three aforementioned sentences to a third person's view, obtains the following:

1. The person had a great time catching up with their friends at the coffee shop.

2. The person finished reading a great book and is looking for their next one.

3. The person biked to the park and watched the birds for an hour.

## Small talk integration into medical information

A program was developed that executed the following procedure on the USMLE Step 3 questions. Through sentence tokenization, each question was broken down into individual sentences, and a small talk sentence was inserted. Once processed, each sentence from the USMLE question was followed by a sentence from the small talk file, creating an alternating sequence, as shown in Fig 1.

The final dataset included a total of 488 questions: 122 multiple-choice questions and 122 open-ended questions, each presented with and without small talk.

## ChatGPT queries

ChatGPT was prompted using the OpenAI API (in Python). Each query was submitted separately as a new query, i.e., our program read each question from the file and submitted it to

> A 37-year-old man comes to the emergency department because he has felt nauseated and light-headed for the past hour. The person is finally getting the hang of a new coding language. Medical history is significant for esophageal varices secondary to alcohol-related cirrhosis and ascites treated with spironolactone. The person went for a walk but forgot to take their wallet with them. He drinks eight to ten alcoholic beverages daily. The person listened to their mom and tried to understand what she was trying to say. While you are obtaining additional history, the patient vomits a large volume of bright red blood and becomes difficult to arouse. The person watched the sunset over the lake. Vital signs are temperature $36.0°C$ ($96.8°F$), pulse 110/min, respirations 12/min, and blood pressure 90/50 mm Hg. The person is planning to go to the movies with friends, eager to see a new Marvel movie. Following initiation of intravenous fluids, what is the most appropriate immediate management?

**Fig 1. Example of a question from the open-ended question dataset with added small talk sentences.** The small talk sentences, added for this illustration, are highlighted in green (the actual dataset does not contain any color highlighting).

ChatGPT. We used the openai.ChatCompletion.create a function with the default parameters https://openai.com/blog/openai-api. The full set of questions was submitted as a user query without a system message for both versions of ChatGPT. In addition, the questions, including small talk, were submitted to ChatGPT-3.5 using the following system message: "You will be asked a question that may contain some irrelevant information. You must first write all the relevant information, then reason about the person's medical condition, and only then attempt to answer the question." We refer to the version with the system message as ChatGPT-3.5 ST-Identify.

## ChatGPT answers assessment

All the responses from ChatGPT to the various datasets were evaluated by a single board-certified physician (MS). For both multiple-choice and open-ended formats, ChatGPT's responses were validated against the official answers of the original multiple-choice questions.

## Statistical analysis

The study investigates the impact of small talk mixed with medical data on the accuracy of medical advice provided by ChatGPT, comparing its performance between versions 3.5 and 4. The primary outcome measures were the accuracy of responses to USMLE Step 3 questions, both multiple-choice and open-ended, with and without small talk sentences included. The study employed a mixed-model design, incorporating both within-subject and between-subject factors. The primary independent variable was the presence or absence of small talk sentences added to the USMLE Step 3 questions, while the version of ChatGPT (3.5 vs. 4) served as a between-subject factor. For each question type (multiple-choice and open-ended), the accuracy of ChatGPT's responses with and without small talk sentences was compared using statistical tests appropriate for the data distribution and study design. Specifically, paired t-tests or Wilcoxon signed-rank tests were conducted to assess within-subject differences in accuracy between conditions (with vs. without small talk sentences) for each ChatGPT version. Additionally, independent t-tests, or Mann-Whitney U tests were used to compare the accuracy of ChatGPT-3.5 and ChatGPT-4 across conditions. Descriptive statistics, including

means, standard deviations, medians, and interquartile ranges, were reported to summarize the accuracy of ChatGPT's responses in each condition and for each version. Inferential statistics, such as p-values, were provided to determine the significance of differences observed. The statistical analyses were performed using Python (version 3.10.1) with the Scipy library (version 1.10.1). The chi2_contingency function from the scipy. Stats library was utilized to compare different groups, and P values of less than 0.05 were considered statistically significant.

## Results

The overall performance of ChatGPT-4 was significantly better than ChatGPT-3.5, with an overall of 61.7% of correct responses vs. 75.4% respectively (p. value <0.001). A significantly better score was observed for ChatGPT-4 version when comparing the overall answer to the USMLE question without the addition of small talk (75.4% vs 66.8%, p.value = 0.045) and (75.4% vs 56.6%, p. value <0.001) for the question including small talk addition (Fig 2). In addition, the effect of small talk integration within medical information differs between the two ChatGPT versions. ChatGPT-3.5, showed a clear decrease in the answers' accuracy when small talk sentences were added to the medical data, with a significant decrease of 66.8% to 56.6% for all ChatGPT-3.5 answers (p. value = 0.025).

While looking at each separate data set of questions, the influence of small talk integration on each type of question is more prominent. ChatGPT-3.5 demonstrates a non-significant

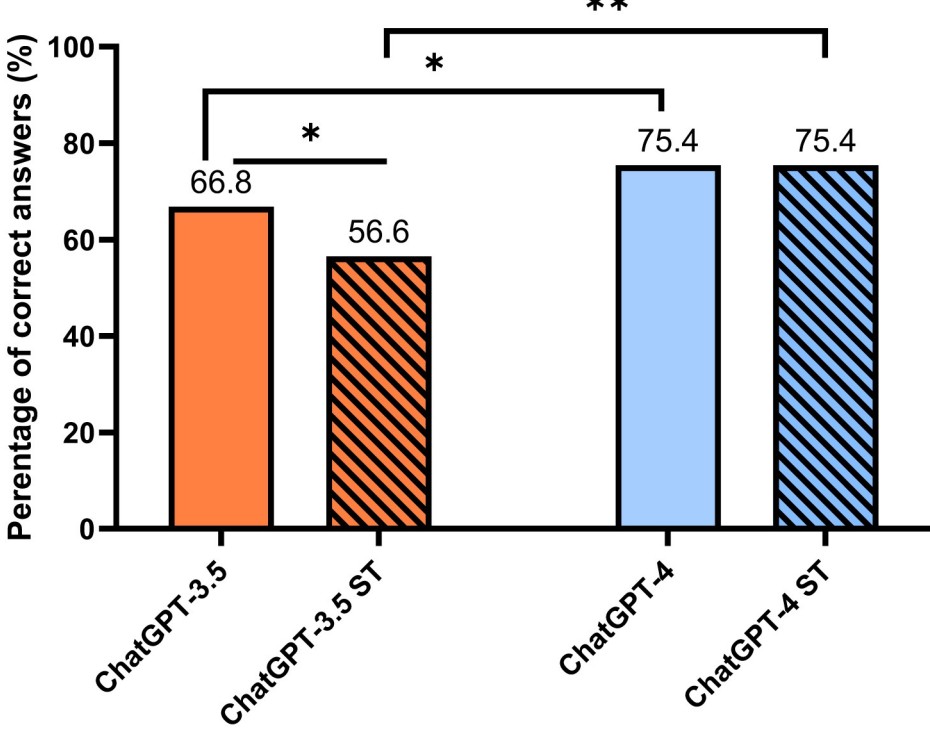

**Fig 2. Result of the performance of ChatGPT-3.5 and ChatGPT-4 on USMLE datasets with and without small talk.** The figure shows the significant difference in ChatGPT-3.5 and ChatGPT-4 performances' with and without the addition of small talk sentences. In addition, it demonstrates the significant difference in the performance of ChatGPT-3.5 for the datasets with and without small talk addition. ST—Small talk, with the addition of small talk to the original question. * and ** indicate statistical significance at levels $p < 0.05$ and $p < 0.001$, respectively.

reduction from 72.1% to 68.9% for the multiple-choice questions, while a more considerable and significant drop in performance from 61.5% to 44.3% (p. value = 0.01) was observed for open-ended questions. In contrast, the performance of ChatGPT-4 remained unchanged despite the introduction of small talk, displaying 67.2% and 83.6% of correct answers for open and multiple-choice questions, respectively (Fig 3).

Upon closer examination of the answers of ChatGPT to each question, a pattern of error can be observed in ChatGPT responses' when the correct answer is that no further test or investigation was required. For instance, each dataset included two questions, with the correct answer being "No further evaluation is necessary" or "No additional study is indicated." Both ChatGPT versions responded incorrectly in the case of open questions suggesting further investigation or treatment regardless in small talk addition. In contrast, when prompted with the dataset of multiple-choice of questions, ChatGPT 3.5 answered one of the 2 questions right when no small talk was inserted, and was disturbed by the addition of small talk and responded wrong for both questions after adding small talk. ChatGPT-4 also improved its score on multiple questions and got one correct answer. In contradiction to ChatGPT3.5, its answer was not impaired by adding small talk, and the performance was the same even after adding irrelevant information.

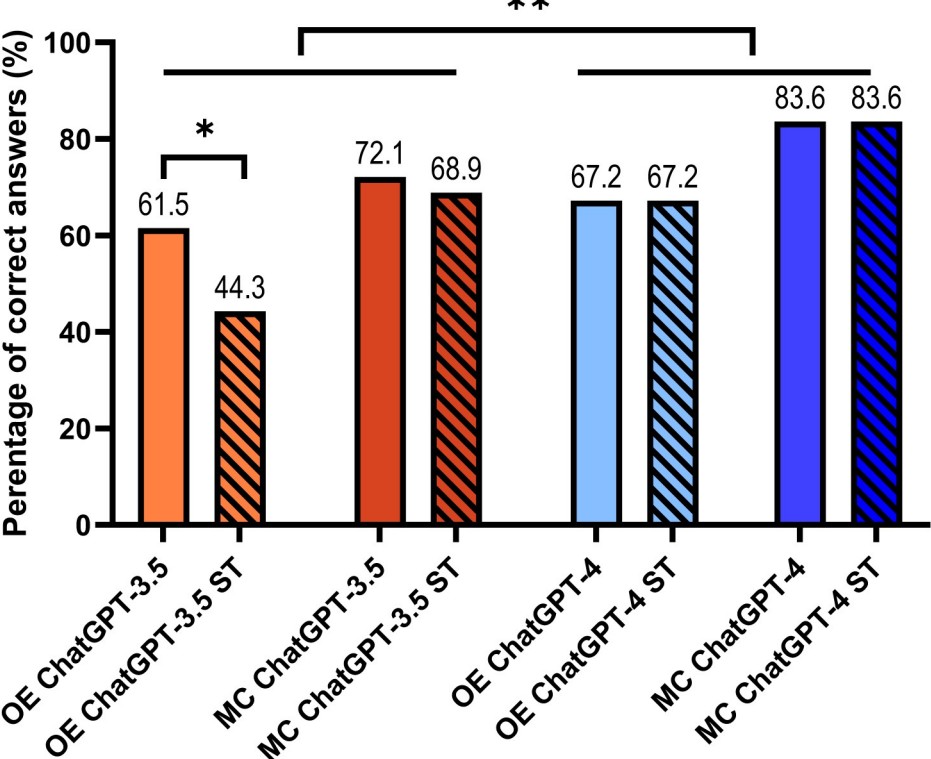

**Fig 3. Performance of ChatGPT-3.5 and ChatGPT-4 on the two types of USMLE questions, i.e., multiple-choice and open-ended, with and without small talk.** ChatGPT-4 performed significantly better than ChatGPT-3.5 (p<0.001). The small talk seemed to have a larger effect on the performance of ChatGPT-3.5 in the open-ended questions. ST—Small talk, with the addition of small talk to the original question. OE—Open-ended questions, MC—Multiple-choice questions. * and ** indicate statistical significance at levels $p < 0.05$ and $p < 0.001$, respectively.

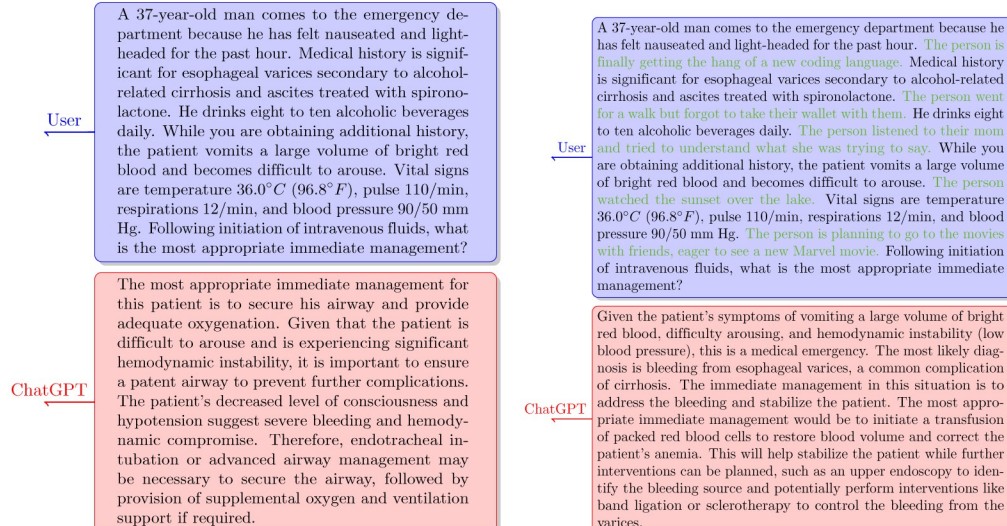

**Fig 4. Comparison of ChatGPT-3.5 responses to a question with and without small talk.** (a) ChatGPT-3.5 provides a correct answer to a question that does not include small talk interference. (b) ChatGPT-3.5 incorrectly responds to a question mixed with small talk (highlighted in green).

In other questions, where the correct answer was a diagnosis or treatment, the addition of small talk impaired ChatGPT-3.5 performance. For example, as seen in Fig 4a, the response was correct before adding small talk. However, as shown in Fig 4b, once small talk phrases were added to the question, ChatGPT-3.5 failed and provided an incorrect response. Interestingly, even though the small talk caused ChatGPT to respond incorrectly, it does not explicitly mention any of the small talk information in its answer and does not explain its wrong answer based on the specific interference added to this question.

Finally, we compare ChatGPT-3.5 ST (without a system message) with ChatGPT-3.5 ST-I-dentify, which contained a system message encouraging ChatGPT-3.5 first to identify the important information and only then answer. The system message did not improve the overall performance of ChatGPT-3.5 on the questions with small talk (p = 0.577). While the performance on the open-ended questions slightly increased from 44.3% to 50.0%, the performance on the multiple-choice questions decreased from 68.9% to 62.3%, with an average performance of 56.1%.

## Discussion

The primary purpose of this study was to investigate the effect of the addition of small talk to medical data on the accuracy of medical advice provided by ChatGPT. First, as expected, ChatGPT-4 outperforms ChatGPT-3.5 with an overall higher score for open and multiple-choice questions. This matches the expectation as ChatGPT-4 is a more advanced version and has been shown to surpass ChatGPT-3.5 on multiple-choice questions in the US and Japan medical exams [21, 37]. However, this is the first study to show a similar improvement in the capacity of ChaGPT-4 to surpass ChatGTP-3.5, giving medical recommendations to open questions that simulate daily clinical needs. The high score of almost three-quarters of correct answers of ChatGPT-4 for open questions in our study indicates its ability to process medical information. These findings suggest the capacity of ChatGPT-4 to respond and provide medical advice and demonstrate its potential future use in the medical field.

When evaluating the effect of small talk addition to the different datasets, ChatGPT-3.5 showed a slight drop in performance for multiple-choice questions and a significant one in answering the open-ended questions following the addition of small talk. In contrast, ChatGPT-4's performance was consistent regardless of small talk, with stable accuracy rates for both question types. To our knowledge, this is the first study evaluating the effect of small talk on ChatGPT and other LLMs' efficacy in processing medical information in the context of unformal or irrelevant information. Our study demonstrates the various impacts of adding small talk on different versions of ChatGPT. It implies that the addition of small talk does not impair ChatGPT-4 performance in processing medical data, which can provide the same accuracy in medical recommendations as in 'medical only' conversation. During a provider-patient interaction, irrelevant information is often mixed with medical data, which needs to be processed and summarized in contrast to the small talk. It has been demonstrated in a previous study that ChatGPT can summarize and provide a note for 'medical only' physician–patient encounters [23]. Therefore, our data suggests that ChatGPT-4 can assist in this task without being impaired by a patient-provider casual discussion that might occur and be provided to ChatGPT in a transcript. These findings provide important answers for medical practitioners and LLM developers regarding the potential of the implication of ChatGPT and other LLMs as a tool in medicine. This is especially important as it is predicted that chatbots will be used by medical professionals, as well as by patients, with increasing frequency [23].

The analysis of the exact scoring of ChatGPT in our study demonstrates that ChatGPT-3.5 answered 72.1% of the multiple-choice questions correctly without small talk integration. This score is higher than the one reported by Kung et al. [37] ranging from 68.8% to 61.5%. It should be noted, however, that our study was conducted approximately 8 months after the original assessment. A possible explanation for this difference is that ChatGPT, as an Artificial Intelligence system, has learned and adapted from the data. As it encounters more information, it refines its models, which often leads to improved performance and accuracy [39]. It is plausible that the elevated scores observed in our research can be attributed to a marked learning enhancement. These findings likely underscore the continuous improvement of ChatGPT over time. We are optimistic that subsequent studies will yield even more favorable outcomes, enhancing ChatGPT's ability to offer even better medical recommendations and furnish dependable support to healthcare providers in medical record documentation.

Each dataset included 2 questions, where the correct answer was that no further investigation was required. Both versions of ChatGPT answers to the open-ended questions were wrong. In contrast, for multiple-choice questions, ChatGPT3.5 had one of the two questions answered correctly if no small talk was added and both were wrong after this addition, whereas ChatGPT-4 was not influenced by small talk addition constantly answers to one of two questions correctly. Our study is the first to report the need and complexity of LLMs to respond to those types of questions. These types of answers are crucial in medicine as patients can be easily referred to countless further tests and investigations, burdening the patients and the medical system [40]. These queries challenge LLMs for whom specific wording of the prompt influence dramatically the answer provided [41]. In those examples, asking what should be the next step may induce that a next step is indeed required. That finding demonstrates the complexity of using ChatGPT in different queries and the need to acknowledge the limit of this technology at this current development.

Finally, we seek to analyze the cause of the small talk disturbance to ChatGPT-3.5 processing. We hypothesized that adding different subjects and specific words would engender a failure in the process of ChatGPT-3.5. However, while the presence of small talk impaired the performance of ChatGPT-3.5 for the datasets of questions, the answer provided by ChatGPT-3.5 did not explain the wrong answer based on a specific subject or word included in the small

talk. This result is concerning, as by delivering incorrect responses but still not mentioning any unrelated information, it may be difficult for a health provider reviewing the answers to pinpoint errors. A sub-analysis of ChatGPT-3.5 did not show significant differences when assessing a different prompt, indicating the irrelevance of some information. This is noteworthy, as previous studies suggested varying prompts could yield different responses in ChatGPT. However, due to the endless options of possible prompts, it's essential to be cautious about generalizing these findings, even though they strengthen our results. We recommend exploring different prompts across various versions to assess their impact.

Therefore, according to these findings, the study suggests that ChatGPT3.5 should not be utilized by practitioners for charting and summarizing health care patient interaction. Additionally, our findings suggests that LLM's have the capacity to streamline the note-taking process for health practitioners to enable doctors to have more time to assess each case and provide medical advice. However, it is important to note that our result also reflected the errors made by ChatGPT in both versions. Therefore, a health practitioner should always be involved as ChatGPT cannot be used independently.

Our study has several limitations. The prominent one is that it is challenging to mimic the small talk that is occurring between the health provider and a patient. In our model, we framed the small talk in the context of "talking to a friend" rather than a physician to avoid bias and integration of medical terms. However, in practice, the patient will be talking to a physician; thus, even the small talk may resemble medical information being conveyed. Such small talk might deteriorate the performance of ChatGPT3.5 and might even affect the performance of ChatGPT-4, which, in our analysis, seemed immune to small talk.

To elucidate the differences in communication context—specifically between sentences someone might express to a friend versus those they might share with their physician—we conducted a supplementary survey using Mechanical Turk. In this survey, we asked 10 participants to produce 5 sentences each that they would likely convey to their physician. We then utilized BERT embeddings [42] to assess the average cosine similarity between pairs of sentences: one from the USMLE original questions and the other from our collection of 'small talk' sentences. This yielded a similarity value of 0.6451. Subsequently, we compared the USMLE questions cosine similarity with the original small talk sentences, obtaining a value of 0.6007. For perspective, we used ChatGPT-4 to generate 50 random sentences and measured their similarity with the USMLE questions. This produced a result of 0.5620. This demonstrates that sentences spoken in a medical context are more semantically related to the USMLE questions than those from casual conversations or random utterances. This distinction emphasizes the need to recognize the unique characteristics of different conversational contexts and the risks of drawing broad conclusions without considering these nuances.

In addition, in our work, the small talk sentences and the medical information were added in an alternating sequence to USMLE questions, each small talk sentence was added as a standalone piece of information. However, in medical practice, the transcript of physician-patient interaction may be much longer than a USMLE question, and the small talk might be structured differently. USMLE question has been used previously to assess medical data processing [37], reinforcing the use of a dataset for such a purpose and allowing us to compare our results. Nevertheless, it is possible that different patterns of small talk integration on different scripts might have various effects on ChatGPT's ability to provide medical counsel. We would also like to stress that this work focuses on both medical information and small talk conveyed in text; however, in practice, the irrelevant information can be conveyed in different modes, such as images (either medical-related images or pictures of the patient's family, pets, etc.) or sounds (either caused by a medical condition of the patient, or the patient laughing as a response to a joke, imitating their boss, etc.). Despite this, the present analysis provides

important new information about the impact of the most common way of communicating [19, 26], including irrelevant information, in physician–patient encounters on the ability of the different versions of ChatGPT to provide medical advice.

Another potential limitation of this study is that it focuses on ChatGPT-only and has not assessed different LLMs and therefore cannot be generalized to other forms of LLMs. Future research could thus attempt to investigate whether the addition of small talk interferes with other LLMs (such as BERT, Cloude, LLAMA-1, and LLAMA-2) ability to provide medical advice.

In this paper, we took the first step toward understanding the performance of the two ChatGPT versions, when faced with physician-patient interactions including medical mixed with irrelevant information. Those unique interactions raised a challenge to discern the impact of casual conversations on the accuracy and reliability of medical recommendations made by these LLMs. This analysis shows that while ChatGPT-3.5 performance was significantly impaired by small talk addition, ChatGPT-4 performance was not affected. The results demonstrate that for some LLMs (i.e., ChatGPT-4 in our case) adding casual conversation does not impair medical advice or diagnosis made by them. Therefore, some LLMs can potentially be used to generate clinical notes from a written transcript. Today's technology has already automated the conversion of audio transcription to written transcript in real-time. Combining these technologies can reduce the time healthcare workers invest in generating medical notes. However, LLM developers, and especially healthcare providers, must be aware of the limitations present in other LLMs (ChatGPT-3.5 in our case), that do not perform well once clinical information is mixed with casual conversation.

## Acknowledgments

This work was supported, in part, but the Ministry of Science and Technology, Israel.

## Author Contributions

**Conceptualization:** Amos Azaria.

**Data curation:** Amos Azaria.

**Methodology:** Myriam Safrai, Amos Azaria.

**Software:** Amos Azaria.

**Supervision:** Myriam Safrai, Amos Azaria.

**Writing – original draft:** Myriam Safrai, Amos Azaria.

**Writing – review & editing:** Myriam Safrai, Amos Azaria.

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
