## [Decision Letter · Decision Letter 0]

27 Oct 2023

PONE-D-23-29585Does small talk with a medical provider affect ChatGPT's medical counsel? Performance of ChatGPT on USMLE with and without distractionsPLOS ONE

Dear Dr. Azaria,

Thank you for submitting your manuscript to PLOS ONE. After careful consideration, we feel that it has merit but does not fully meet PLOS ONE’s publication criteria as it currently stands. Therefore, we invite you to submit a revised version of the manuscript that addresses the points raised during the review process.

We look forward to receiving your revised manuscript.

Kind regards,

Fadi Aljamaan

Academic Editor

PLOS ONE

Journal Requirements:

Additional Editor Comments:

Thanks for addressing this interesting topic especially with the widespread use of AI chatbots like ChatGPT by public. Please address the reviewers' comments and revise your manuscript. 

Reviewers' comments:

Reviewer's Responses to Questions

**Comments to the Author**

1. Is the manuscript technically sound, and do the data support the conclusions?

Reviewer #1: Yes

Reviewer #2: Yes

2. Has the statistical analysis been performed appropriately and rigorously? 

Reviewer #1: I Don't Know

Reviewer #2: Yes

3. Have the authors made all data underlying the findings in their manuscript fully available?

Reviewer #1: Yes

Reviewer #2: Yes

4. Is the manuscript presented in an intelligible fashion and written in standard English?

Reviewer #1: Yes

Reviewer #2: Yes

5. Review Comments to the Author

Reviewer #1: First and foremost, I commend the authors for pioneering research on the impact and efficacy of ChatGPT in the context of medical reasoning during healthcare-patient interactions, particularly when interspersed with casual conversations. The chosen methodology, modeling of data, and the analytical processes presented in the manuscript are technically robust and indeed constitute a solid scientific subject of study. The blending of casual interactions with medical discourse represents a novel avenue of exploration and the insights presented are valuable.

That being said, I would like to put forth a few comments that go beyond the current scope of the study but could be considered as avenues for further enhancement or future research directions:

1 Small Talk Sentences: The decision to procure small talk sentences from the Mechanical Turk platform is an interesting one. However, it would be valuable to dive deeper into the semantic and contextual relationship of these casual statements with genuine medical information. I suggest a quantitative analysis of the meaning of these casual sentences and their numerical similarities with the to-be-inserted medical data. This can be achieved by embedding these long and complex sentences into vectors, allowing for a richer, multi-dimensional analysis. Additionally, this method can set the stage for a more extensive quantitative study on varying amounts and structures of casual conversation insertions, gauging their impact on the overall interaction.

2 Importance of In-Context Learning: The innate ability of ChatGPT for in-context learning is a pivotal aspect of its operation. Minor variations in the phrasing or structure of question prompts could elicit significantly different outputs from the model. I believe that meticulous crafting of these prompts, considering this nuance, has the potential to notably augment GPT-4's accuracy rates. It would be a promising topic to delve into in subsequent studies. At the very least, I recommend the authors to touch upon this crucial aspect in the discussion section of the paper, shedding light on the importance of prompt design and its potential implications for the efficacy of the model in medical interactions.

In conclusion, the presented research is a significant stride in the right direction, offering valuable insights into the blending of casual discourse with medical reasoning in physician-patient interactions using LLMs. I believe that the considerations mentioned above can further enhance the depth and scope of this research.

Reviewer #2: I really enjoyed this paper, it was easy to read and I don't think it needs revisions.

My only tiny comment is in the abstract you say there is a difference for C-GPT 3.5, but actually only one of the two comparisons is significant, and I wonder if this makes sense.

I also think you should discuss with the journal how best to present your findings. If Figs 4 and 5 could be presented next to each other with the key differences in the results/diagnosis, I think it would look pretty.

Your discussion highlights lots of areas of further work and I really hope you are able to do this. As you highlight, the small talk in a consultation might be about medical issues, possibly a family members or something long since solved, and it would be good to try these too.

Congratulations on this paper.

6. PLOS authors have the option to publish the peer review history of their article (what does this mean?). If published, this will include your full peer review and any attached files.

Reviewer #1: **Yes: **Qiyang Hu

Reviewer #2: No

---

## [Author Response · Author response to Decision Letter 0]

10 Nov 2023

The following is a copy-paste from the response letter. It is highly recommended to read it from the word version of the file.

Editor-in-Chief

Re: PONE-D-23-29585 revision

Dear Editor, 

We are pleased to submit our revised manuscript entitled: " Does small talk with a medical provider affect ChatGPT’s medical counsel? Performance of ChatGPT on USMLE with

and without distractions" for further consideration for publication in PLOS ONE. 

Thank you for the editorial consideration and careful reviews of our manuscript. 

We have addressed the reviewers' helpful comments and are now submitting a revised manuscript. We are grateful for the remarks and suggestions and feel that the current manuscript, incorporating the revisions in response to the comments, is indeed improved. Please find below our point-by-point responses to the reviewers' comments. We appreciate the opportunity to revise our manuscript and hope that our revised manuscript will be found suitable for publication.

Sincerely,

Myriam Safrai MD.

Amos Azaria PhD.

 

Reviewers' 1 comments:

Reviewer: "Small Talk Sentences: The decision to procure small talk sentences from the Mechanical Turk platform is an interesting one. However, it would be valuable to dive deeper into the semantic and contextual relationship of these casual statements with genuine medical information. I suggest a quantitative analysis of the meaning of these casual sentences and their numerical similarities with the to-be-inserted medical data. This can be achieved by embedding these long and complex sentences into vectors, allowing for a richer, multi-dimensional analysis. Additionally, this method can set the stage for a more extensive quantitative study on varying amounts and structures of casual conversation insertions, gauging their impact on the overall interaction".

Our response: We thank the reviewer for this comment. As the reviewer suggested, we conducted a quantitative analysis of the similarity between the small talk and the medical data. In our study, we aimed to simulate nonmedical small talk by asking the participants in our survey to provide 5 different sentences as they were talking to a friend. Following the reviewer's comment, we conducted a supplementary Mechanical Turk survey. In this survey, we asked 10 participants to produce 5 sentences each that they would likely convey to their physician. We then utilized BERT embeddings \\cite{devlin2018bert} to assess the average cosine similarity between pairs of sentences: one from the USMLE original questions and the other from our collection of ' small talk' sentences. This yielded a similarity value of 0.645. Subsequently, we compared the USMLE question's cosine similarity with the original small talk sentences, obtaining a value of 0.601. This analysis indeed demonstrates that our original small talk is more general. In addition, we used ChatGPT-4 to generate 50 random sentences and measured their similarity with the USMLE questions. This produced a result of 0.5620. Subsequently, in the discussion, we emphasize the need to recognize the unique characteristics of different conversational contexts and the risks of drawing broad conclusions without considering these nuances. In our revised manuscript, we have added a paragraph following our limitation section (lines 275-289):" To elucidate the differences in communication context—specifically between sentences someone might express to a friend versus those they might share with their physician—we conducted a supplementary survey using Mechanical Turk. In this survey, we asked 10 participants to produce 5 sentences each that they would likely convey to their physician. We then utilized BERT embeddings [41] to assess the average cosine similarity between pairs of sentences: one from the USMLE original questions and the other from our collection of' small talk' sentences. This yielded a similarity value of 0.6451. Subsequently, we compared the USMLE questions' cosine similarity with the original small talk sentences, obtaining a value of 0.6007. For perspective, we used ChatGPT-4 to generate 50 random sentences and measure their similarity with the USMLE questions. This produced a result of 0.5620. This demonstrates that sentences spoken in a medical context are more semantically related to the USMLE questions than those from casual conversations or random utterances. This distinction emphasizes the need to recognize the unique characteristics of different conversational contexts and the risks of drawing broad conclusions without considering these nuances".

Reviewer: "Importance of In-Context Learning: The innate ability of ChatGPT for in-context learning is a pivotal aspect of its operation. Minor variations in the phrasing or structure of question prompts could elicit significantly different outputs from the model. I believe that meticulous crafting of these prompts, considering this nuance, has the potential to notably augment GPT-4's accuracy rates. It would be a promising topic to delve into in subsequent studies. At the very least, I recommend the authors to touch upon this crucial aspect in the discussion section of the paper, shedding light on the importance of prompt design and its potential implications for the efficacy of the model in medical interactions”.

 Our response: We agree with the reviewer. When conducting our initial study, the full set of questions was submitted as a user query without a system message for both versions of ChatGPT. In addition, the questions that included small talk were submitted to ChatGPT-3.5 using the following system message prompt, which was phrased with an attempt to assist ChatGPT-3.5 in identifying important information: “You will be asked a question that may contain some irrelevant information. You must first write all the relevant information, then reason about the person's medical condition, and only then attempt to answer the question.'' No significant difference was found between the performance for both prompts (p=0.577). We have added the description in the methods section: “The full set of questions was submitted as a user query without a system message for both versions of ChatGPT. In addition, the questions, including small talk, were submitted to ChatGPT-3.5 using the following system message: “You will be asked a question that may contain some irrelevant information. You must first write all the relevant information, then reason about the person's medical condition, and only then attempt to answer the question.” We refer to the version with the system message as ChatGPT-3.5 ST-Identify.” (lines 129-135). We presented the result in the appropriate section (lines 185-191): “Finally, we compare ChatGPT-3.5 ST (without a system message) with ChatGPT-3.5 ST-Identify, which contained a system message encouraging ChatGPT-3.5 first to identify the important information and only then answer. The system message did not improve the overall performance of ChatGPT-3.5 on the questions with small talk. While the performance on the open-ended questions slightly increased from 44.3% to 50.0%, the performance on the multiple choice questions decreased from 68.9% to 62.3%, with an average performance of 56.1%.” 

We discussed our result in the discussion section and recommend pursuing more evaluation with different probes (lines 261-266): “ChatGPT-3.5 did not show significant differences when assessing a different prompt, indicating the irrelevance of some information. This is noteworthy, as previous studies suggested varying prompts could yield different responses in ChatGPT. However, due to the endless options of possible prompts, it’s essential to be cautious about generalizing these findings, even though they strengthen our results. We recommend exploring different prompts across various versions to assess their impact.”.

Reviewer 2

1- Reviewer: "I really enjoyed this paper, it was easy to read and I don't think it needs revisions. My only tiny comment is in the abstract you say there is a difference for C-GPT 3.5, but actually only one of the two comparisons is significant, and I wonder if this makes sense".

Our response: We thank the reviewer for pointing this out. We have clarified this in our abstract: The analysis results demonstrate that the ability of ChatGPT-3.5 to answer correctly was impaired when small talk was added to medical data (66.8% vs. 56.6%; p=0.025). Specifically, for multiple-choice questions (72.1% vs. 68.9%; p=0.67) and for the open questions (61.5% vs. 44.3%; p=0.01), respectively”.

2- Reviewer: " I also think you should discuss with the journal how best to present your findings. If Figs 4 and 5 could be presented next to each other with the key differences in the results/diagnosis, I think it would look pretty".

Our response: We agree with the reviewer and have made this change in the manuscript and created for Figure 4 a panel “a” and “b.”. If the journal wants to discuss the data display, we will gladly do so. However, we think that following this comment and this adaptation, the data is well presented.

---

## [Decision Letter · Decision Letter 1]

27 Feb 2024

PONE-D-23-29585R1Does small talk with a medical provider affect ChatGPT's medical counsel? Performance of ChatGPT on USMLE with and without distractionsPLOS ONE

Dear Dr. Azaria,

Thank you for submitting your manuscript to PLOS ONE. After careful consideration, we feel that the manuscript is well structured and it is almost ready for publication. However, one reviewer raised some minor points that, if addressed, might further boost the attractiveness of your paper. The comments are minor and not much time should be required to address them. We invite you to submit a revised version of the manuscript that addresses the points raised during the review process.

We look forward to receiving your revised manuscript.

Kind regards,

Massimo Stella, PhD

Academic Editor

PLOS ONE

Journal Requirements:

Reviewers' comments:

Reviewer's Responses to Questions

**Comments to the Author**

1. If the authors have adequately addressed your comments raised in a previous round of review and you feel that this manuscript is now acceptable for publication, you may indicate that here to bypass the “Comments to the Author” section, enter your conflict of interest statement in the “Confidential to Editor” section, and submit your "Accept" recommendation.

Reviewer #1: All comments have been addressed

Reviewer #3: (No Response)

2. Is the manuscript technically sound, and do the data support the conclusions?

Reviewer #1: Yes

Reviewer #3: Yes

3. Has the statistical analysis been performed appropriately and rigorously? 

Reviewer #1: I Don't Know

Reviewer #3: Yes

4. Have the authors made all data underlying the findings in their manuscript fully available?

Reviewer #1: Yes

Reviewer #3: Yes

5. Is the manuscript presented in an intelligible fashion and written in standard English?

Reviewer #1: Yes

Reviewer #3: Yes

6. Review Comments to the Author

Reviewer #1: Thank you to the authors for their thorough responses to my earlier comments. The revised paper effectively addresses the concern I raised regarding quantitative semantic and contextual sentence analysis in the original version. It also presents well-balanced discussions on in-context learning, utilizing customized prompting techniques, and maintains an appropriate length. This enhanced research not only opens new avenues but also lays foundational steps for future in-depth exploration in the fascinating field of large language models. The publication of this paper will undoubtedly be a good contribution to both the medical and AI communities, aiding in the understanding and application of ChatGPT across various domains.

Reviewer #3: (No Response)

7. PLOS authors have the option to publish the peer review history of their article (what does this mean?). If published, this will include your full peer review and any attached files.

Reviewer #1: **Yes: **Qiyang Hu

Reviewer #3: No

---

## [Editor Report · Decision Letter 2]

1 Apr 2024

Does small talk with a medical provider affect ChatGPT's medical counsel? Performance of ChatGPT on USMLE with and without distractions

PONE-D-23-29585R2

Dear Dr. Azaria,

We’re pleased to inform you that your manuscript has been judged scientifically suitable for publication and will be formally accepted for publication once it meets all outstanding technical requirements.

Kind regards,

Massimo Stella, PhD

Academic Editor

PLOS ONE